# Epidemiology and referral patterns of patients living with chronic kidney disease in Johannesburg, South Africa: A single centre experience

**Yusuf Urade**[1], **Zaheera Cassimjee**[1], **Chandni Dayal**[1,2], **Sheetal Chiba**[1,2], **Adekunle Ajayi**[1,2], **Malcolm Davies**[1,2]*

1 Department of Internal Medicine, University of the Witwatersrand, Johannesburg, Gauteng, South Africa,
2 Division of Nephrology, Helen Joseph Hospital, Johannesburg, Gauteng, South Africa

* malcolm.davies@wits.ac.za

## Abstract

Chronic kidney disease (CKD) is a significant contributor to the global burden of non-communicable disease. Early intervention may facilitate slowing down of progression of CKD; recognition of at-risk patient groups may improve detection through screening. We retrospectively reviewed the clinical records of 960 patients attending a specialist nephrology outpatient clinic during the period 1 January 2011–31 December 2021. A significant proportion (47.8%) of patients were referred with established CKD stage G4 or G5. Non-national immigration status, previous diagnosis with diabetes, and advancing age were associated with late referral; antecedent diagnosis with HIV reduced the odds of late referral. Black African patients comprised most of the sample cohort and were younger at referral and more frequently female than other ethnicities; non-nationals were younger at referral than South Africans. Hypertension-associated kidney disease was the leading ascribed aetiological factor for CKD (40.7% of cases), followed by diabetic kidney disease (DKD) (19%), glomerular disease (12.5%), and HIV-associated kidney disease (11.8%). Hypertension-related (25.9%) and diabetic (10.7%) kidney diseases were not uncommon in people living with HIV. Advancing age and male sex increased the likelihood of diagnosis with hypertensive nephropathy, DKD and obstructive uropathy; males were additionally at increased risk of HIV-associated kidney disease and nephrotoxin exposure, as were patients of Black African ethnicity. In summary, this data shows that hypertension, diabetes, and HIV remain important aetiological factors in CKD in the South African context. Despite the well-described risk of CKD in these disorders, referral to nephrology services occurs late. Interventions and policy actions targeting at-risk populations are required to improve referral practices.

## Introduction

An increasing global prevalence of CKD [1] mandates that healthcare programmes develop policy and infrastructure to address the growing patient population living with kidney dysfunction [2].

**Data Availability Statement:** All relevant data for this study are available from the University of the

Witwatersrand repository (https://doi.org/10.54223/20.500.12430/551161).

**Funding:** The authors received no specific funding for this work.

**Competing interests:** The authors have declared that no competing interests exist.

The burden on health imposed by CKD is especially prominent in low- and middle-income countries [1, 3], with sub-Saharan Africa carrying a disproportionately high burden of disease [1]. The prevalence of CKD in this region has been estimated to be between 13.9 [4] 17.7% [5], exceeding the reported global average of 9.1% [1]. A complex interplay of genetic and socio-economic factors contribute to the increased risk of CKD development and progression in individuals of Black African ethnicity [6, 7]; accelerated onset of kidney failure (KF) has been reported in South African series [8].

Despite the evident need, access to nephrology care in sub-Saharan Africa remains poor [9, 10]. South Africa boasts one of the few state-funded kidney replacement therapy (KRT) programmes in the region [11], yet treatment access rates for patients living with KF reported for the country are amongst the lowest in the world [12].

Early referral to specialist nephrology services slows down progression of CKD and reduces risk of premature cardiovascular death [13]. Public health initiatives and interventions at the sub-specialist level reduce incident CKD [14], and may thus reduce strain on overburdened nephrologists in resource-limited settings such as sub-Saharan Africa.

Adequate capacitation of nephrology services and development of public health interventions require understanding of local epidemiology of CKD. Interrogation of referral patterns for patients living with CKD may additionally identify areas for improvement in sub-specialist care. Such data remains poorly described for sub-Saharan Africa. We therefore undertook the largest survey of patients diagnosed with CKD in South Africa to date with a view to better characterising CKD epidemiology and referral patterns.

## Methods

### Study population

A retrospective file review of patients attending the Renal Outpatient Clinic at Helen Joseph Hospital (HJH) during the period 1 January 2011–31 December 2021 was undertaken. The population served by the hospital is ethnically and socio-economically diverse. Black Africans make up 69.8% of residents compared to 22.9% for whites, 3.2% for Asians, and 2.9% for those of mixed ancestry; with 34.4% residing in informal housing [15].

### Study site

HJH is a 636-bed state-funded tertiary-level academic facility affiliated with the University the Witwatersrand. The Division of Nephrology at HJH provides specialist nephrology care to communities resident in the northwestern areas of Johannesburg. The divisional medical staff establishment comprises three board-certified specialist nephrologists, two nephrology specialty trainees, and three junior doctors in training for certification as specialist physicians. Nephrology services provided include both diagnostic evaluation (radiological and serological workup, as well as renal biopsy) and management of all aspects of kidney disease, including dialysis services. The weekly renal outpatient clinic undertaken by the Division provides nephrologist-led care to an average of 60 patients per clinic, with an average of 8 new referrals reviewed by a nephrology specialist at each clinic. Referrals are received from a broad spectrum of non-specialist healthcare providers, including nurse-led primary healthcare clinics and general practitioners located in the communities served by HJH, and non-nephrology specialist units within the facility.

### Definitions

Chronic kidney disease was broadly defined in accordance with the 2012 Kidney Disease: Improving Global Outcomes (KDIGO) guidelines [16]; that is, the presence of abnormality of

kidney structure or function present for more than 3 months. To meet this definition, only those patients with documented follow-up of at least 3 months who demonstrated persistence of abnormality were considered for inclusion in this study.

CKD stage at referral was assigned as per eGFR as determined be the Chronic Kidney Disease Epidemiology Collaboration (CKD-EPI) equation and as defined by KDIGO [16].

### Determination of sample size

In-centre clinical activity reports were used to estimate the total number of patients on regular outpatient follow-up over the period of this study at 29040 individuals. Further analysis of clinic review rounds determined the average percentage of patients with documented CKD to be 80% per clinic. At a 95% confidence level and 2.5% margin of error, the representative sample size was thus determined to be 960 patients.

### Data extraction and storage

Simple random sampling using a lottery method was deployed to identify patients meeting study definition of CKD for inclusion in this study. Data pertaining to patient demographics (age at referral, sex, ethnicity, and country of origin); comorbid disease (hypertension, diabetes mellitus, and HIV status); ascribed aetiology of chronic kidney disease as determined by the attending nephrologist; and eGFR on referral were extracted anonymously from clinical records selected using convenience sampling and stored on a secured database prior to analysis using Stata v. 17.0 (StataCorp, College Station, Texas). Clinical records were accessed and data extracted during the period 7/10/2022–26/12/2022.

### Statistical analysis

Age and eGFR at referral were compared between ethnicities, sexes, and nationality categories using Mann Whitney U or Student t testing, as indicated. Stepwise multivariate logistic regression was used to determine the effect of age, gender, and ethnicity on aetiological category of CKD. Stepwise logistic regression modelling was used to evaluate the effect of demographics, immigration status, and comorbidity on referral patterns.

### Ethical considerations

Permission to undertake this study was granted by the Human Research Ethics Committee (HREC) of the University of the Witwatersrand (certificate M220956); the study was undertaken in adherence with the principles of the Declaration of Helsinki. Formal consent to participate in this study was waived by the former in view of the retrospective nature of this study and in consideration of the preservation of data anonymity.

### Results

Demographics, severity of renal dysfunction, comorbidities, and ascribed aetiology of CKD of the sample cohort are shown in Table 1.

Variation in age at referral and sex were observed between ethnicities. Patients of Black African ethnicity were younger at referral than other ethnic groups (54.6 ± 15.4 years and 62.5 ± 15.1 years respectively, p < 0.001); women formed a slight majority of patients in this group (51.3%) whist males were in the majority in non-Black ethnicities (56.4%, p = 0.024). Non-nationals were younger at referral than South African nationals (48.9 ± 12.4 compared to 58.3 ± 15.8 years, p < 0.001); males were over-represented in the former group (60% compared to 50.1%, p = 0.051). Ascribed aetiology of CKD showed variation with demographic

**Table 1. Demographics, referral characteristics, and ascribed aetiology of CKD in sample cohort (n = 960).**

| | |
|---|---|
| *Ethnicity\** | |
| Black African | 641 (66.8%) |
| Caucasian | 127 (13.2%) |
| Mixed ancestry | 117 (12.2%) |
| Asian | 75 (7.8%) |
| *Sex\** | |
| Male | 492 (51.3%) |
| Female | 468 (48.7%) |
| *Age at referral (years)\*\** | 57.2 ± 15.7 |
| *Nationality\** | |
| South African nationals | 850 (88.5%) |
| Non-nationals | 110 (11.5%) |
| *Severity of renal dysfunction on referral* | |
| CKD-EPI eGFR (mL/min/1.73m$^2$)$^+$ | 31 (17–48) |
| CKD stage\* | |
| CKD stage G1 | 84 (8.8%) |
| CKD stage G2 | 50 (5.2%) |
| CKD stage G3a | 141 (14.7%) |
| CKD stage G3b | 226 (23.5%) |
| CKD stage G4 | 261 (27.2%) |
| CKD stage G5 | 198 (20.6%) |
| *Comorbidities* | |
| Hypertension | 701 (73.0%) |
| HIV | 309 (32.2%) |
| Diabetes mellitus | 293 (30.5%) |
| *Ascribed aetiology of CKD$^+$* | |
| Hypertension-associated kidney disease | 391 (40.7%) |
| Diabetic kidney disease | 182 (19.0%) |
| Glomerular diseases | 120 (12.5%) |
| HIV-associated kidney diseases | 113 (11.8%) |
| Obstructive uropathy | 64 (6.7%) |
| Nephrotoxin exposure$^{++}$ | 31 (3.2%) |
| Autosomal dominant polycystic kidney disease | 31 (3.2%) |
| Previous documented acute kidney injury$^§$ | 15 (1.6%) |
| Congenital anomalies of genitourinary tract | 8 (0.2%) |
| Renovascular disease | 2 (0.2%) |
| Inherited tubulopathy | 2 (0.2%) |
| Cardiorenal syndrome | 1 (0.1%) |

\*n (% of total N),

\*\*mean ± SD,

$^+$median (interquartile range), +as determined by attending nephrologist,

$^{++}$Nephrotoxin exposure comprises 28 cases of nonsteroidal anti-inflammatory nephropathy and 3 cases of tenofovir disoproxil fumarate chronic nephropathy with no other aetiology apparent,

$^§$Previous documented acute kidney injury (AKI) includes patients known with previous episodes of AKI with residual renal dysfunction and comprises 10 cases of sepsis-associated AKI, 4 cases of rhabdomyolysis, and 1 case of the HELLP (hemolysis, liver dysfunction, and low platelets) syndrome.

**Table 2. Effect of demographic parameters on ascribed aetiology of CKD.**

| | | *Relative risk (95% CI)* | *p* |
|---|---|---|---|
| Age | Obstructive uropathy | 1.05 (1.01–1.08) | 0.004 |
| | Diabetic kidney disease | 1.04 (1.02–1.07) | <0.001 |
| | Hypertension-associated kidney disease | 1.03 (1.00–1.06) | 0.022 |
| | Glomerular disease | 0.94 (0.92–0.97) | 0.002 |
| Male sex | Obstructive uropathy | 11.04 (4.04–30.22) | <0.001 |
| | Nephrotoxin exposure | 3.56 (1.14–9.85) | 0.027 |
| | Hypertension-associated kidney disease | 3.51 (1.52–8.08) | 0.003 |
| | Diabetic kidney disease | 3.48 (1.47–8.25) | 0.005 |
| | HIV-associated kidney diseases | 3.14 (1.29–7.67) | 0.012 |
| Black African ethnicity | HIV-associated kidney diseases | 5.91 (2.29–15.24) | <0.001 |
| | Nephrotoxin exposure | 3.35 (1.07–10.50) | 0.038 |

parameters. Increasing age was associated with higher likelihood of diagnosis with hypertension-associated kidney disease, diabetic kidney disease (DKD), and obstructive uropathy, but reduced the risk of diagnosis with glomerular disease; male sex was associated with increased risk of hypertension-associated kidney disease, DKD, HIV-associated kidney disease, obstructive uropathy, and nephrotoxin exposure. Black African ethnicity increased the risk of diagnosis with HIV-associated kidney diseases and CKD due to nephrotoxin exposure (Table 2).

The 309 persons living with HIV (PLWH) included in the sample were more frequently of Black African than other ethnicity (88.0% compared to 12.0% for other ethnicities, p < 0.001); women contributed a larger proportion of this cohort than did males (55.7% versus 44.3%, p = 0.003) (Table 3).

A significant proportion of patients (47.8%) were referred with established CKD of stage G4 or G5. Increasing age, non-South African nationality, and diagnosis with comorbid diabetes mellitus increased the odds of referral with an eGFR below 30 mL/min/1.73m$^2$; a trend towards later referral was also noted for males (p = 0.053). Conversely, PLWH were less likely to be referred with established CKD of lower eGFR (Table 4). Non-South African nationality remained a significant predictor of late referral when analysis was restricted to patients presenting in CKD stage G5 (OR 2.23, 95% CI 1.44–3.44, p < 0.001).

## Discussion

Although accumulating data evidences high prevalence rates of CKD in South Africa and the sub-Saharan region [4, 5, 17–19], little is known about disease characteristics and referral patterns in patients living with the condition. The present study confirms significant contribution from well-described risk factors including hypertension, diabetes mellitus, HIV infection, and advancing age to the development of CKD in the local context. Despite these known associations, patient referral at early, actionable stages of CKD appears to be poor, with older patients, diabetics, and non-nationals being particularly at risk of late referral.

Up to 48% of the South African population may be hypertensive [20]; the prevalence rate for type 2 diabetes mellitus has been estimated at 15% [21]. South Africa remains the global epicentre of the HIV pandemic with 13% of the population living with the virus [15]. Local studies have reported CKD prevalence rates of 21% in PLWH [22], 18% in diabetics [23], and 25% in hypertensive patients [24], although considerable variation exists dependant on the definition of CKD used. The diagnosis of hypertension-associated kidney disease in a

**Table 3. Demographics, referral characteristics, and ascribed aetiology of CKD in persons living with HIV (n = 309).**

| | |
|---|---|
| *Ethnicity** | |
| Black African | 272 (88.0%) |
| Mixed ancestry | 29 (9.4%) |
| Caucasian | 6 (1.9%) |
| Asian | 2 (0.6%) |
| *Sex** | |
| Male | 137 (44.3%) |
| Female | 172 (55.7%) |
| *Age at referral (years)*** | 54.0 ± 16.9 |
| *Nationality** | |
| South African nationals | 270 (87.4%) |
| Non-nationals | 39 (12.6%) |
| *Severity of renal dysfunction on referral* | |
| CKD-EPI eGFR (mL/min/1.73m$^2$)$^+$ | 37.7 ± 24.1 |
| CKD stage* | |
| CKD stage G1 | 15 (4.9%) |
| CKD stage G2 | 15 (4.9%) |
| CKD stage G3a | 69 (22.3%) |
| CKD stage G3b | 89 (28.8%) |
| CKD stage G4 | 74 (23.9%) |
| CKD stage G5 | 47 (15.2%) |
| *Comorbidities* | |
| Hypertension | 176 (57.0%) |
| Diabetes mellitus | 40 (12.9%) |
| *Ascribed aetiology of CKD* | |
| HIV-associated kidney diseases | 113 (36.6%) |
| Hypertension-associated kidney disease | 80 (25.9%) |
| Diabetic kidney disease in the setting of HIV | 33 (10.7%) |
| Glomerular diseases in the setting of HIV | 24 (7.8%) |
| Obstructive uropathy | 10 (3.2%) |
| Previous documented acute kidney injury | 8 (2.6%) |
| Autosomal polycystic kidney disease | 4 (1.3%) |
| Nephrotoxin exposure | 3 (1.0%) |
| Tenofovir nephropathy | 2 (0.6%) |
| Congenital anomalies of genitourinary tract | 2 (0.6%) |
| Inherited tubulopathy | 1 (0.3%) |

*n (% of total N),

**mean ± SD,

$^+$median (interquartile range)

predominantly Black African population is, however, controversial [25]. Whilst hypertension is a known contributor to the progression of CKD, the presence of elevated blood pressure may in these patients evidence underlying genetic or developmental pathophysiologies such as *APOL1*-related kidney disease [26] or reduced nephron endowment [27].

Rapid urbanization and significant structural socio-economic inequalities in the local context [28] conspire to create a milieu favourable to the progression of CKD [29]. Historical

**Table 4. Logistic regression, predictors of referral in CKD stages G4 or G5.**

|  | *OR (95% CI)* | *p* |
|---|---|---|
| Age (years) | 1.01 (1.00–1.02) | <0.001 |
| Non-South African nationality | 2.05 (1.35–3.12) | <0.001 |
| Comorbid diabetes mellitus | 1.45 (1.08–1.96) | 0.015 |
| Comorbid HIV infection | 0.71 (0.53–0.95) | 0.019 |
| Black African ethnicity | 0.83 (0.62–1.13) | 0.230 |
| Male sex | 1.29 (0.99–1.68) | 0.053 |

injustices render patients of Black African ethnicity most likely to experience these risk factors, which contributes to younger presentation with advanced CKD as observed in this study.

Whilst CKD prevalence is generally higher in women than in men, males are known to exhibit more rapid deterioration in renal function; this may account for the slight preponderance of males and the trend towards increased odds of presentation in CKD G4 or G5 observed for men in this study [30]. Male sex is known to accelerate progression of DKD to KF, an association which may reflect the effect of androgenic hormones on inflammatory and fibrotic pathways [31]. Adequate treatment of hypertension, a key factor in slowing progression of CKD, appears to be more difficult to attain in males [32]. Several factors may underlie this apparent sex discrepancy in blood pressure control. Poorer self-reported adherence to antihypertensive drugs amongst male respondents has been noted in some studies [33], although similar rates of antihypertensive prescription refill between sexes in others [34] may evidence biological differences in hypertension pathways [32]. These factors may underlie the increased risk of DKD and hypertension-associated kidney disease in males observed in this study.

People living with HIV comprised nearly one-third of referrals to nephrology services in this cohort, almost three times the estimated prevalence rate of HIV in the broader South African population [15]. PLWH included in this study were more commonly female and of Black African ethnicity, mirroring the demographics of HIV infection in the general population where a complex interplay of biological infection factors coupled with historically entrenched racial and gender inequalities render young Black African women most at risk of infection [35]. HIV incidence rates in South Africa are highest in the 15–29-year-old age group, whilst prevalence rates are highest in women aged 30–49 years [36]. PLWH in the present study tended to be older, reflective of improved survival in the era of universal access to antiretroviral therapy (ART) [37]. Non-communicable diseases, including hypertension and diabetes, have been shown to be increasing in PLWH as that population ages [38]. Age-related changes in the disease profile of PLWH are appreciable in the present study where hypertension-associated kidney disease and DKD are important contributors to CKD in these patients.

Immigrants constitute an at-risk group for CKD; migrants from sub-Saharan Africa are especially at risk due to a combination of genetic, environmental, and healthcare system-related factors in their respective countries of origin [10, 39]. South African legal restrictions on the employment of non-nationals compel immigrants to seek unprotected employment in the informal sector, often as day labourers and at reduced wages; the country's crumbling bureaucracy hinders formalization of residency status which might otherwise permit escape from these conditions [40]. The excessive physical labour and restrictions on standards of living which result are likely to accelerate CKD progression. This may explain the increasing risk of CKD with duration of emigration reported in other studies [41]. Difficulties in accessing the South African state healthcare system due to lack of immigration documentation and systemic

xenophobia [42] may additionally have contributed to the tendency for referral with advanced CKD exhibited by non-nationals in the present study.

Overall, referral with advanced CKD was not uncommon in this cohort with nearly half of patients presenting with established CKD stage G4 or G5. Late referral is a long-established referral is a long-established challenge to improving dialysis-free survival the world over: on average, 30% of patients living with kidney failure are under the care of a nephrologist for less than 6 months before dialysis initiation [43]. Such late referrals appear more frequent in the developing world, possibly reflecting poorer access to specialist nephrology services. In India and Brazil, which share BRICS organization membership with South Africa, late referrals account for 52 and 58% of dialysis initiations, respectively [44, 45]. Data from the African context is lacking, but previous reports suggest that 50% of CKD patients in Nigeria manifest indications for urgent dialysis initiation at referral [46]. Late referral in such resource-constrained settings has immediate implications for patient outcomes: South African data suggests that less than a third of patients presenting with CKD G5 requiring dialysis will be initiated onto this life-saving therapy [47, 48]. Referral at earlier, actionable stages of CKD is commensurably poor. Overall, only 3% of Danish patients with CKD G3 –G5 are referred for regular nephrology follow-up; rates of referral increase in proportion to CKD stage with 35% of CKD G5 patients undertaking regular follow-up comparted to 11% of CKD G4 patients [49]. Previous African studies have reported high rates of advanced CKD at referral to nephrology services, with 57% of patients in a Cameroonian study presenting with CKD G4 or G5 [50], and 97% of Nigerian patients presenting with similar levels of renal dysfunction [46].

Older age, non-Caucasian ethnicity, presence of comorbidities, and lack of medical insurance have been associated with increased probability of late referral [43]. Local data supports a lack of health system responsiveness at the primary care level for aging patients, especially in the state sector, which delays referral to specialist services such as nephrology [51].

Increased risk of referral with advanced stage CKD in diabetics in this cohort is of particular concern. Local data suggests an increased probability of exclusion from state KRT programmes in diabetics due to cardiovascular comorbidity [48] which may have contributed to ennui in referral to nephrology. The advent of sodium-glucose 2 transporter (SGLT2) inhibitors offers the opportunity to substantially retard DKD progression and ameliorate the development of cardiovascular disease burden, with greater benefit being derived from earlier intervention facilitated by referral at lower CKD stages [52]. The experience of kidney disease-targeting interventions in HIV offers a potential roadmap to improve referral of diabetics in the local setting, as evidenced by reduced odds of referral at advanced CKD stage amongst PLWH in this study.

Taken together, these data shed light on current challenges in the management of kidney disease in South Africa. The tendency of the burden of CKD to fall upon younger Black South Africans dependant on the resource-limited state sector mandates that healthcare systems and policy should be developed to reduce late referral to nephrology services. Trends towards increased risk of exposure to nephrotoxins in this CKD-vulnerable population require public educational interventions. Referral systems for diabetics require urgent review to reduce late presentation and subsequent exclusion from KRT programmes. Whilst referral of PLWH does not appear to be subject to these limitations, improved survival of these patients in the era of universal access to ART is likely to increase demand on nephrology services as the prevalence of hypertension-associated kidney disease and DKD rise in this population. Evaluation of trends in the nephrologist workforce in South Africa show that the country lacks sufficient specialists to provide adequate care for the existing CKD population; expansion of nephrologist training programmes is required to meet the anticipated increase in demand [53]. Late presentation of non-nationals with advanced CKD is an additional cause for concern. Lack of

nephrology services in their countries of origin and financial limitations on the ability of these immigrants to seek care elsewhere render these patients dependant on the South African state-funded healthcare system [11]. South Africa at present lacks clear guidelines on the provision of medical care to non-nationals living with CKD [11]; policy needs to be developed to assist this comparatively small and vulnerable population without further compromising access to care for South Africans.

There are limitations to this study. The single-centre nature of this work, and the urbanised cohort recruited, may limit generalizability to the broader South African and sub-Saharan context. Use of simple random sampling to recruit patients may represent an additional source of bias through the inclusion of widely dispersed data. Substantial reliance on clinical, rather than histological, diagnoses for non-glomerular disease may have resulted in over-representation of hypertension-associated kidney disease as an ascribed aetiological factor. The lack of albuminuria assay on referral limits full assessment of CKD stage as recommended by KDIGO guidelines [15]. However, since referrals to this unit characteristically rely on eGFR as the primary indication, this limitation does not materially compromise analysis of referral patterns.

## Conclusions

Hypertension, diabetes, and HIV remain important aetiological factors in CKD in South Africa. Despite the recognised association of these comorbidities with CKD, many patients are referred late in disease course with advanced kidney dysfunction. Rapid progression of CKD in patients of Black African ethnicity results in younger presentation with important ramifications for kidney health systems. Non-nationals constitute an at-risk population for the development of CKD; these patients are particularly vulnerable to delays in referral. Evidence from the referral of people living with HIV suggests that specialist interventions may improve early referral; improvements in referral of diabetic patients are especially required, and public education initiatives to reduce nephrotoxin exposure need to be considered. Finally, healthcare policy for non-nationals is urgently required to improve provision of nephrology care to these patients.

## Acknowledgments

The authors gratefully acknowledge the contribution of the Division of Nephrology at Helen Joseph Hospital nursing and administrative staff in accessing clinical records for data collection.

## Author Contributions

**Conceptualization:** Yusuf Urade, Chandni Dayal, Malcolm Davies.

**Data curation:** Yusuf Urade, Malcolm Davies.

**Formal analysis:** Malcolm Davies.

**Investigation:** Yusuf Urade, Zaheera Cassimjee, Malcolm Davies.

**Methodology:** Yusuf Urade, Zaheera Cassimjee, Malcolm Davies.

**Project administration:** Zaheera Cassimjee, Malcolm Davies.

**Supervision:** Malcolm Davies.

**Writing – original draft:** Yusuf Urade, Zaheera Cassimjee, Malcolm Davies.

**Writing – review & editing:** Yusuf Urade, Zaheera Cassimjee, Chandni Dayal, Sheetal Chiba, Adekunle Ajayi, Malcolm Davies.

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
