## [Decision Letter · Decision Letter 0]

4 Dec 2023

PGPH-D-23-01737

Epidemiology and referral patterns of patients living with chronic kidney disease in Johannesburg, South Africa: a single centre experience

Dear Dr. Davies,

Thank you for submitting your manuscript to PLOS Global Public Health. After careful consideration, we feel that it has merit but does not fully meet PLOS Global Public Health’s publication criteria as it currently stands. Therefore, we invite you to submit a revised version of the manuscript that addresses the points raised during the review process.

We look forward to receiving your revised manuscript.

Kind regards,

Valerie Ann Luyckx

Academic Editor

Journal Requirements:

Additional Editor Comments (if provided):

The paper has been reviewers by 2 global experts on chronic kidney disease. The topic is important and such studies are needed. It is however important to understand more about how the patients were selected for inclusion. A sample size calculation was given, please explain more how the actual charts were then retrieved for inclusion here to ensure adequate representation of the clinic population. Were these all patients or a subset of patients? it is unclear where the total number given over the 10 years was patients or visits.

In addition, the reviewers comments are outlined below and should be addressed prior to re-submission.

Reviewers' comments:

Reviewer's Responses to Questions

**Comments to the Author**

1. Does this manuscript meet PLOS Global Public Health’s publication criteria? Is the manuscript technically sound, and do the data support the conclusions? The manuscript must describe methodologically and ethically rigorous research with conclusions that are appropriately drawn based on the data presented.

Reviewer #1: Yes

Reviewer #2: Yes

2. Has the statistical analysis been performed appropriately and rigorously?

Reviewer #1: Yes

Reviewer #2: Yes

3. Have the authors made all data underlying the findings in their manuscript fully available (please refer to the Data Availability Statement at the start of the manuscript PDF file)?

Reviewer #1: Yes

Reviewer #2: Yes

4. Is the manuscript presented in an intelligible fashion and written in standard English?

Reviewer #1: Yes

Reviewer #2: Yes

5. Review Comments to the Author

Reviewer #1: General comment

1. Well done to the authors for all the efforts to produce this manuscript and to publish

2. The referral patterns are needed to put in public health interventions for early referrals as there are measures to prevent progression.

3. It would be helpful to know how referral diagnosis differ from nephrologist diagnosis.

4. Effort should be made to report the proportions of late referrals and how they compare to other regions in Africa and globally.

5. It will help to report which cadre of health workforce do refer to the facility of the authors.

6. Some more details required for the study setting.

Title

7. Did the nephrologist agree to the referral diagnosis or are the diagnosis used the diagnosis of the attending nephrologist?

Abstract

8. Focus on late referrals and those who are referred with kidney failure should take centre stage

Introduction

9. I suggest authors do not start the write up with NCDs and I think it might be helpful to go straight to CKD and explore justification for the study based on what is known.

10. The burden on health imposed by CKD is especially prominent in low- and middle-income countries, [3],[5] with sub-Saharan Africa carrying a disproportionate burden of disease. is this high or low burden? Should come out clearer.

11. The prevalence of CKD in this region has been estimated to be 17.7%, [6]. I think it will be helpful to quote Stanifer et al 2014 also as a complete study than a subgroup analysis?

Methods

12. Some information from the hospital or study setting would be helpful – number of beds, number of patients per clinic, how many OPD days, number of nephrologists etc. which cadre of health workforce refer to the unit? Medical officers, nurses, specialist or even nephrologist? More information on study setting and referral patterns recommended

13. Why is this in inverted commas? “Chronic kidney disease” was broadly defined in accordance with the 2012 Kidney Disease: Improving Global Outcomes (KDIGO) guidelines.

14. The sample size was determined to be 960. How were these selected? what sampling method was used? This may introduce a lot of bias in the dataset.

Results

15. Not sure how the categorization of the other regions in table 1 would be useful. Can others keep it simple as nationals and non-nationals?

16. What proportion of patients required renal replacement on admission? Authors should seek to discuss this with regards to other studies in Africa and globally

17. Ascribed etiology in table 1 is this the nephrologists diagnosis or referral diagnosis? Would help to know how referral diagnosis differ from nephrologist diagnosis in the teaching hospital.

18. Some diagnosis in table 1 quite vague – what is nephrotoxins exposure or previous documented AKI? what is the real etiology?

19. “women formed a slight majority of patients in this group (51.3%) whist males were in the majority in non-Black ethnicities (56.4%, p = 0.024).” sentences like these should be well written to show comparison and read well.

20. A lot of sentences are repetition from the tables. Authors can save word count by referring to the table.

21. From table 3 About 30 patients were referred with stage 1 and 2 CKD. was wodering the indication for referral with normal renal function to the nephrologist. Nephrotic syndrome? what are the proportions of these then?

22. Was table 4 determined by logistic regression? Why is the 6th column of Table 4 empty?

Discussion

23. the presence of elevated blood pressure may in these patients evidence underlying genetic or developmental pathophysiology such as APOL1-related kidney disease or reduced nephron endowment. This statement should be referenced

24. Whilst CKD prevalence is generally higher in women than in men, males are known to exhibit more rapid deterioration in renal function; this may account for the slight preponderance of males in this study. [29]. This statement can be justified if more males presented with CKD4 or 5 in your study. Is this the case. Then it helps to show.

25. Adequate control of hypertension appears more difficult to attain in males, [31]. is compliance not a factor. Fear of erectile dysfunction as side effects of anti hypertensives etc?

26. PLWH included in this study were more commonly female and of Black African ethnicity, mirroring the demographics of HIV infection in the general population. HIV incidence rates in South Africa are highest in the 15 – 29 year-old age group, Whilst prevalence rates are highest in women aged 30 – 49 years[32]. Any plausible reasons for this?

27. Late referral is a long-established barrier to improving dialysis-free survival the world over, with 30% of dialysis patients lacking significant pre-initiation nephrologist follow-up. [39]. It will help to compare late referral to others in the sub-region or Africa.

28. The advent of sodium-glucose 2 transporter (SGLT2) inhibitors offers the opportunity to substantially retard DKD progression and ameliorate the development of cardiovascular disease burden, with greater benefit being derived from earlier intervention facilitated by referral at lower CKD stages. [42]. Is this for only DKD? how well is SGLT2 used in SA? Are uptakes low and what may be the reasons for low uptakes?

Reviewer #2: Congratulation's! Is a very well- designed article. I've only made some suggestions. It is very important to publish this to make the world know about the reality of your country as a manner to change it.

6. PLOS authors have the option to publish the peer review history of their article (what does this mean?). If published, this will include your full peer review and any attached files.

**Do you want your identity to be public for this peer review?** For information about this choice, including consent withdrawal, please see our Privacy Policy.

Reviewer #1: No

Reviewer #2: **Yes: **María Carlota González- Bedat

---

## [Decision Letter · Decision Letter 1]

26 Mar 2024

Epidemiology and referral patterns of patients living with chronic kidney disease in Johannesburg, South Africa: a single centre experience

PGPH-D-23-01737R1

Dear Dr Davies,

We are pleased to inform you that your manuscript 'Epidemiology and referral patterns of patients living with chronic kidney disease in Johannesburg, South Africa: a single centre experience' has been provisionally accepted for publication in PLOS Global Public Health.

Best regards,

Valerie Ann Luyckx

Academic Editor

Reviewer Comments (if any, and for reference):

Reviewer's Responses to Questions

**Comments to the Author**

1. If the authors have adequately addressed your comments raised in a previous round of review and you feel that this manuscript is now acceptable for publication, you may indicate that here to bypass the “Comments to the Author” section, enter your conflict of interest statement in the “Confidential to Editor” section, and submit your "Accept" recommendation.

Reviewer #1: All comments have been addressed

Reviewer #3: All comments have been addressed

2. Does this manuscript meet PLOS Global Public Health’s publication criteria? Is the manuscript technically sound, and do the data support the conclusions? The manuscript must describe methodologically and ethically rigorous research with conclusions that are appropriately drawn based on the data presented.

Reviewer #1: Yes

Reviewer #3: Yes

3. Has the statistical analysis been performed appropriately and rigorously?

Reviewer #1: Yes

Reviewer #3: N/A

4. Have the authors made all data underlying the findings in their manuscript fully available (please refer to the Data Availability Statement at the start of the manuscript PDF file)?

Reviewer #1: Yes

Reviewer #3: Yes

5. Is the manuscript presented in an intelligible fashion and written in standard English?

Reviewer #1: Yes

Reviewer #3: Yes

6. Review Comments to the Author

Reviewer #1: Thanks for response to the queries.

I have no further comments.

Reviewer #3: I've evaluated that the authors have addressed my concerns and made the

manuscript acceptable for publication.

7. PLOS authors have the option to publish the peer review history of their article (what does this mean?). If published, this will include your full peer review and any attached files.

**Do you want your identity to be public for this peer review?** For information about this choice, including consent withdrawal, please see our Privacy Policy.

Reviewer #1: No

Reviewer #3: No
